

# Elevational surveys of Sulawesi herpetofauna 1: Gunung Galang, Gunung Dako Nature Reserve

Benjamin R. Karin[1,*], Isaac W. Krone[1,*], Jeffrey Frederick[1,2], Amir Hamidy[3], Wahyu Tri Laksono[3], Sina S. Amini[1], Evy Arida[4], Umilaela Arifin[1,5], Bryan H. Bach[6], Collin Bos[1,7], Charlotte K. Jennings[1], Awal Riyanto[3], Simon G. Scarpetta[1], Alexander L. Stubbs[1] and Jimmy A. McGuire[1]

[1] Museum of Vertebrate Zoology and Department of Integrative Biology, University of California, Berkeley, CA, United States
[2] Field Museum of Natural History, Chicago, IL, United States
[3] Research Center for Biosystematics, Badan Riset dan Inovasi Nasional (BRIN), Cibinong, Bogor, Indonesia
[4] Research Center for Applied Zoology, Badan Riset dan Inovasi Nasional (BRIN), Cibinong, Bogor, Indonesia
[5] Center for Taxonomy and Morphology, Leibniz Institute for the Analysis of Biodiversity Change, Universität Hamburg, Hamburg, Hamburg, Germany
[6] California Institute for Quantitative Biosciences (QB3), University of California, Berkeley, Berkeley, CA, United States
[7] Division of Evolutionary Biology, Faculty of Biology II, Ludwig-Maximilians-Universität München, Planegg-Martinsried, Germany
* These authors contributed equally to this work.

Corresponding author
Benjamin R. Karin,
benkarin@berkeley.edu

## ABSTRACT

The Indonesian island of Sulawesi has a unique geology and geography, which have produced an astoundingly diverse and endemic flora and fauna and a fascinating biogeographic history. Much biodiversity research has focused on the regional endemism in the island's Central Core and on its four peninsulas, but the biodiversity of the island's many upland regions is still poorly understood for most taxa, including amphibians and reptiles. Here, we report the first of several planned full-mountain checklists from a series of herpetological surveys of Sulawesi's mountains conducted by our team. In more than 3 weeks of work on Gunung Galang, a 2,254 m peak west of the city of Tolitoli, Sulawesi Tengah Province, on Sulawesi's Northern Peninsula, we recovered nearly fifty species of reptiles and amphibians, more than a dozen of which are either new to science or known but undescribed. The incompleteness of our sampling suggests that many more species remain to be discovered on and around this mountain.

## INTRODUCTION

Sulawesi is a global hotspot of species endemism, yet the biodiversity of the island is still poorly known and documented (*Koch, 2012*). Sulawesi's montane areas have been

particularly undersampled and are likely to harbor immense biodiversity as do other tropical mountains (*Rahbek et al., 2019b*). A clear understanding of the biodiversity on Sulawesi is essential for understanding conservation priorities and necessary measures for the incredible montane ecosystems it harbors. In this study, we summarize the herpetological results of a large-scale biodiversity survey undertaken in 2018 by a sizable team of scientists over several weeks on Gunung Galang, a peak located within the Gunung Dako Forest Reserve and adjacent to the peak for which the reserve is named, Gunung Dako. This expedition was part of a multi-year project to survey the biodiversity of mountains across Sulawesi.

The island of Sulawesi formed from the accretion of several paleo-islands over the past several million years, and the Northern Peninsula alone is expected to have been separated into four or more paleo-islands (*Nugraha & Hall, 2018*). This geographic history has likely acted as a primary driver of species and population level divergences across the island, and accounts for species boundaries in several species groups (*Evans et al., 2003*; *Frantz et al., 2018*; *McGuire et al., 2023*). Species restricted to higher elevation environments on Sulawesi have been subject to even greater isolation, separated by shallow marine waters during the rapid uplift and exhumation of igneous bodies that compose the mountains of the southern portion of the northern peninsula (SNP), and later by lowlands as paleo-islands coalesced into the modern landmass.

Few areas were uplifted or exhumed faster than the land comprising and surrounding what is now Toli-Toli Regency. The Northern Peninsula of Sulawesi is geologically young, rising from beneath the ocean mostly in the past 3 Ma, and fusing with Central Sulawesi during the late Miocene (*Hall, 2012*) or early Pleistocene (*Nugraha & Hall, 2018*). The mountains of the SNP are partly composed of the Lalos-Toli pluton, which is one of several late Neogene intrusive bodies in Northwest Sulawesi. The Lalos-Toli pluton occurs east and northeast of Tolitoli, and is located mostly inland from the modern coast. The pluton was dated to the late Miocene at $8.2 \pm 0.2$ Ma (*Maulana et al., 2016*; *Advokaat et al., 2017*), and emplaced at a depth of 11.3 km (*Maulana et al., 2019*). The remarkably rapid exhumation of the Lalos-Toli pluton and adjacent intrusive bodies (*i.e.*, the Sony pluton) can be attributed to activity on the Palu-Koro Fault Zone and other faults of northwest Sulawesi, and the Lalos-Toli pluton was probably exhumed during the late Miocene through the Pliocene (*Maulana et al., 2019*). A paleogeographic reconstruction of Sulawesi indicated that parts of the mountains in the SNP (likely including the Lalos-Toli pluton) breached sea level and uplifted to an elevation of between 500–2,000 m by 3 Ma, during the middle Pliocene (*Nugraha & Hall, 2018*). Further, bathymetric modeling indicates a greater land area during glacial events, such as the Last Glacial Maximum (*Nugraha & Hall, 2018*). Given increased land area coupled with cooler climate regimes of Pleistocene glacial periods, montane connectivity and potential for dispersal likely fluctuated on Sulawesi during the past 2.6 million years. In sum, geologic and bathymetric evidence indicate that the formation of the modern mountain biota of the Northern Peninsula occurred during the late Neogene and Pleistocene.

By contrast, rocks from lower elevations of the SNP mountains and much of the surrounding area are from the Tinombo formation. The age of the Tinombo Formation is

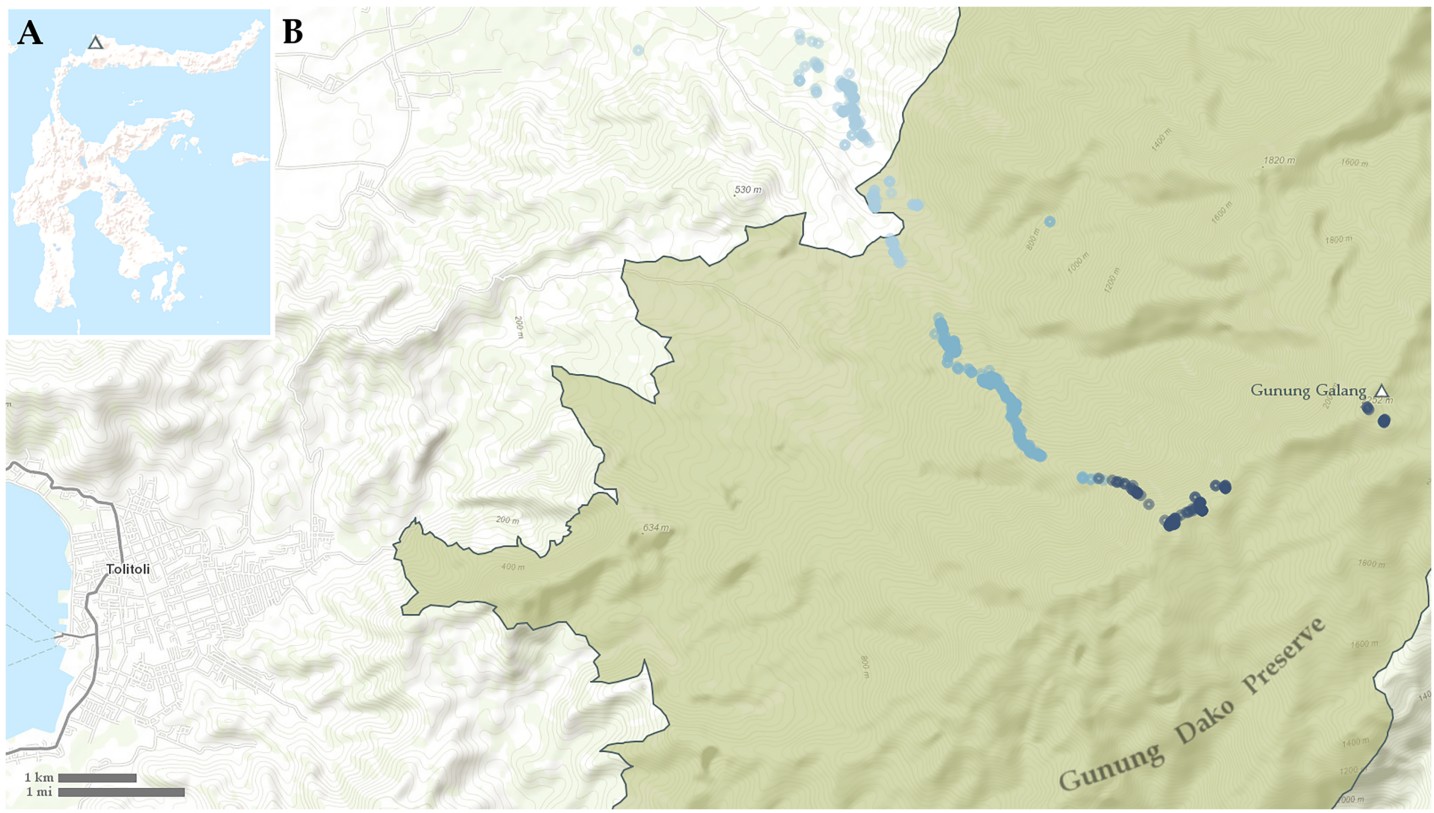

**Figure 1 Maps showing (A) location of Gunung Galang on Sulawesi and (B) detail of Tolitoli and the Western Gunung Dako Preserve with sampling points indicated.** Points less than 700 m elevation in light blue, points from 700–1,400 m elevation in medium blue, points above 1,400 m elevation in dark blue.

poorly constrained, but most age estimates indicate an Eocene age through, at the latest, the earliest Miocene (*van Leeuwen & Muhardjo, 2005*; *Advokaat et al., 2017*). Unlike the mountain biotas, the assemblies of lower elevation biotas do not seem to be intrinsically tied to the development of specific intrusive igneous bodies.

Among the highest of Northern Sulawesi's mountains is Gunung Galang, a 2,254 m a.s.l. peak West of the city of Tolitoli, located within the 19.73 kha Gunung Dako Forest Reserve (Fig. 1). Gunung Galang is part of a relatively small mountain range on the Northern Peninsula that is partially separated from the main crest of the Northern Peninsula by intervening lower elevation hills. The montane habitats of Gunung Galang are connected along the main crest of the peninsula to Gunung Sojol (2,890 m) and Ibuyule Malino (2,410 m), to the Southwest. The mountain range extends South until a low elevation break along the neck of the Northern Peninsula near Magapa (~400 m) and to the east until a narrow break at the Buol fault, followed by a more significant low-elevation break further east at the Gorontalo fault (see Fig. 1 of *Nugraha & Hall, 2018*). The peak's height, location within a protected area, and position within this small, disconnected mountain range make it an enticing candidate for biological study.

In order to survey its potentially unique herpetofauna in detail, we conducted surveys out of two field camps that we established at 1,000 and 1,700 m elevation, as well as the

village of Kinapasan at 300 m elevation. Based out of the higher camp, we routinely conducted surveys at the summit of Gunung Galang. Species from Gunung Galang restricted to elevations above 1,000 m may not be able to easily traverse warmer lowland areas, and moreover, would have been isolated on a separate paleo-island until a few million years ago. We therefore expect montane endemics from this survey to be restricted to one or more of the ranges outlined above, and not necessarily extend further east on the Northern Peninsula than Gorontalo or extend south into the Central Core of Sulawesi.

To our knowledge, there have been no intensive herpetological surveys at higher elevations on the Northern Peninsula before our expedition. Previous sampling by members of our team was not focused on sampling high elevation environments. Pre-1940's expeditions were mostly focused on mammal and bird collections, and only intermittently collected herpetological specimens. This herpetological survey is therefore likely to be the most thorough high elevation expedition to the Northern Peninsula ever conducted.

## MATERIALS AND METHODS

### Field surveys

This expedition was part of a large collaborative project between several institutions including the National Research and Innovation Agency (BRIN, formerly the Indonesian Institute of Sciences, LIPI, including many researchers from Museum Zoologicum Bogoriense (MZB)), Indonesia; Tadulako University, Indonesia; Museum of Vertebrate Zoology (MVZ), USA; American Museum of Natural History (AMNH), USA; and Museum Victoria (MV), USA. Our team consisted of five herpetologists, Jimmy McGuire (MVZ), Awal Riyanto (BRIN), Wahyu Tri Laksono (BRIN), Jeffrey Frederick (MVZ), and Benjamin Karin (MVZ). The mammalogy team consisted of Anang Achmadi (BRIN), Dede Avandi (BRIN), Kevin Rowe (MV), and Heru Handika (MV). The ornithology team consisted of Tri Haryoko (BRIN), Hadi Wikanta (BRIN), Rauri Bowie (MVZ), and Karen Rowe (MV). The ichthyology team consisted of Sopian Sauri (BRIN). The arthropod and invertebrate team consisted of Sarino San (BRIN), Pungki Lupiyaningdya (BRIN), Syahfitri (BRIN), Nyoman Sumerta (BRIN), Fahri (Tadulako University), Peter Oboyski (UC Berkeley), and Anna Holmquist (UC Berkeley). Haemosporidian blood parasites were sampled by Rachael Joakim (AMNH). In total there were 22 scientists and several local guides conducting fieldwork on this expedition. The animal study protocol was approved by the Institutional Animal Care and Use Committee of the University of California Berkeley (AUP-2014-12-6954-2). Research and collection permits were granted by LIPI and RISTEKDIKTI (now BRIN) (23//SI/MZB/VIII/2018).

Our surveys were conducted from 3–27 July 2018. Our work out of the 1,000 and 1,700 m camps was conducted from 3–21 July. On 22 July, we returned to Desa (Village) Kinopasan (320 m elevation) and continued to conduct field surveys in the surrounding habitat until 27 July 2018. Several specimens, including several of the rare snakes we collected, were found by members of the larger expedition team. We performed amphibian and reptile surveys out of our low camp (1,000 m) and high camp (1,700 m), as well as in and around Desa Kinapasan (300 m). We worked primarily along a narrow trail that

followed a ridge to the summit. See photos in Fig. 2 for an overview of different habitats encountered on the mountain.

Our sampling strategies involved daytime and nighttime hand collection at all elevations, drift-fence/pitfall bucket arrays, and sticky traps. Sticky traps (72MAX; Catchmaster, Bayonne, NJ, USA) were stapled in place or laid on the ground where appropriate habitat was present alongside the trail. We used an assortment of placements, including placing them on and beside tree buttresses, directly on large and small diameter tree trunks, and in open leaf litter. Though our traps sampled at regular intervals from 292 m to the summit, we split them into separate lines for convenience, named after the approximate lowest elevation they sampled. The lines were set as follows: Line300 (30 traps from 292–338 m), Line750 (35 traps from 769–939 m), Line1000 (94 traps from 1,000–1,246 m), Line1350 (50 traps from 1,350–1,530 m), Line1500 (50 traps from 1,507–1,659 m), Line1720 (50 traps from 1,720–1,784 m), Line1920 (30 traps from 1,920–1,990 m), Line2050 (30 traps from 2,050–2,090 m), and Line2222 (30 traps from 2,222–2,254 m). Pitfall traps consisted of 20–30 L buckets with associated drift fencing cut from plastic tarps. Pitfall arrays were placed at 314 m (up to 12 buckets, two nights, 24 bucket nights), 1,050–1,075 m (up to 21 buckets, 68 bucket nights), 1,780 m (up to 30 buckets, 98 bucket nights), and 2,170 m (up to11 buckets, 44 bucket nights). There were too few buckets to open all these pitfall traps simultaneously, so buckets were moved between the sites as needed; day-by-day information on pitfall trapping effort is available upon request. We surveyed a total of 222 bucket-nights. Night surveys were conducted each night, usually between 7:00 pm and 1:00 am.

We identified each specimen as accurately as possible in the field. However, many specimens were re-identified later based on physical or molecular analyses, and our identifications in this article reflect our most recent and complete knowledge. Any individuals not referable to described species are referred to here with working names given in quotation marks (*e.g.*, *Limnonectes* sp. "T yellow").

We obtained liver tissue samples in RNAlater for all specimens collected, swabbed all frogs for *Batrachochytrium dendrobatidis* fungus, and took blood smears to screen for *Plasmodium* and other haemosporidian blood parasites for most specimens. Whole stomachs of *Limnonectes* fanged frogs and most scincid lizards were removed and preserved in RNAlater for future gut content analysis.

All specimens collected are housed either at the MZB (half of the specimens, plus most singletons) or at the MVZ. Tissue samples for each specimen were divided with a subsample provided to each collection.

## Statistical analyses

We carried out all analyses in R 4.2.1 (*R Core Team, 2022*). Our code can be downloaded at https://doi.org/10.5281/zenodo.7999977. All of our analyses are based on data from the field catalog of specimens. Elevation data were not recorded for 15 specimens, for which we used the get_elev_point function from the elevatr package (*Hollister et al., 2022*) to estimate their elevations from AWS Open Data Terrain Tiles at a 9.6 m resolution ("*Terrain Tiles, 2022*"). Field-collected elevation data for the 621 other specimens are

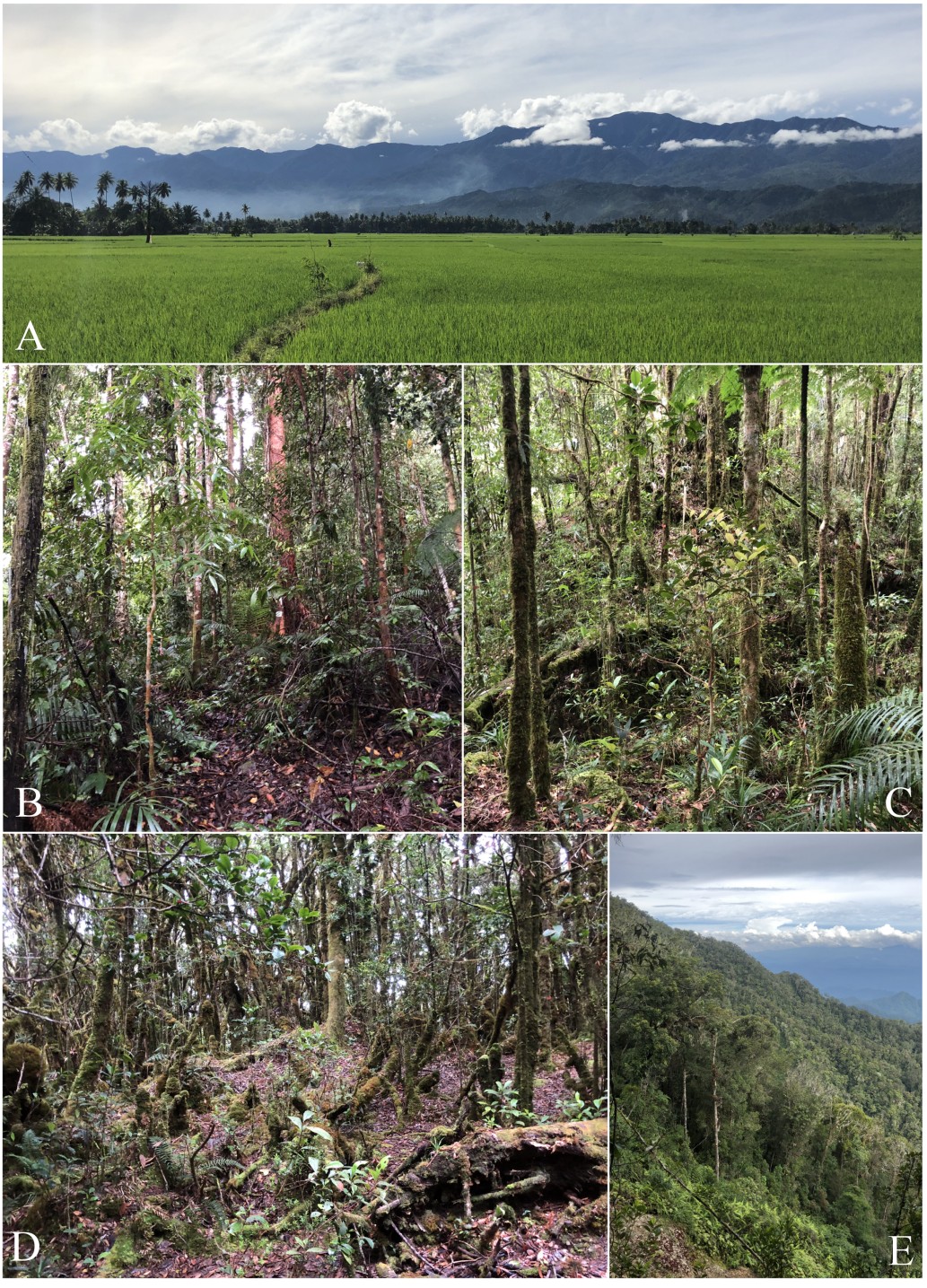

**Figure 2 Photos of habitats present on Gunung Galang.** (A) Rice paddies at ~50 m elevation with views of Gunung Galang and Gunung Dako in the background. (B) Forest at mid camp ~1,100 m. (C) Forest at high camp ~1,650 m. (D) Stunted mossy forest at the summit ~2,250 m. (E) A landslide provided a rare view out of the forest eastwards, photo taken along the ridge near the summit ~2,000 m. Photos by B. Karin.                                                            

highly correlated with elevatr estimates of their elevations ($R^2$ = 0.9989, $p$ < 0.0001). Our elevational species richness curves were produced using the divDyn R package (*Kocsis et al., 2019*), calculating range-through species richness at 10 m intervals.

In order to estimate how many species we did not sample, we used the specpool function from the vegan package (*Oksanen et al., 2022*) to produce Chao (*Chiu et al., 2014*) and second-order Jackknife (*Smith & van Belle, 1984*) species richness estimates based on species abundance data, treating each day of collection as a sample, including a correction term for small-sample size. One specimen, a *Psammodynastes pulverulentus* found at 845 m was excluded from these analyses because its collection date was not recorded.

Terrain tiles for our maps are from ESRI. To understand the risk posed to the herpetofauna by forest loss, we used data on forest loss in Sulawesi (*World Resources Institute, 2022a*), Toli-Toli Regency (*World Resources Institute, 2022b*) and Gunung Dako Preserve (*World Resources Institute, 2022c*) from Global Forest Watch, accessed on October 18th, 2022.

## RESULTS

### Habitat

We observed a substantial difference in the degree of human-disturbance between low and high elevations. The habitat around the village of Kinopasan consisted of mixed kebun (farmland/plantation) primarily of coconut palm intermixed with cacao, banana, and clove trees. Going up in elevation, the farmland transitioned into highly disturbed secondary forest at about 600–700 m elevation. Above 1,000 m, most of our sampling was conducted along a narrow trail through what appeared to be mature forest along a ridge that became narrower and steeper with elevation. We observed a stark transition in habitat when transitioning from the lower-elevation northwest facing ridge to the southwest facing ridge leading to the summit. Immediately after cresting the northwest facing ridge, the forest became much wetter and the trees were covered in a thick layer of moss. This habitat type continued upwards until close to the summit where it transitioned yet again into "mossy forest" or "cloud forest" composed of small stunted trees and vines, pitcher plants, and even thicker moss covering most exposed surfaces of trees and rocks.

The only above-ground streams with visible flowing water were at lower elevations. We surveyed a relatively large stream (the stream bed was ~15 m in width) and smaller tributaries in the immediate vicinity of Kinopasan village. The large stream was flanked on the north side by mixed secondary forest and on the south by farmland. Some members of our team conducted a brief survey up this stream up to ~880 m elevation to a waterfall (Lat = 1.07816, Lon = 120.90411) where we collected only a few specimens. Above this elevation along the main trail, we did not find any significant above-ground flowing streams. In several locations while searching for flowing streams we encountered exposed boulders rather than soil, and could hear the sound of running water several meters below ground. We hypothesize that water may flow above ground during flood events that prevent accumulation of soil in these areas, but that the stream is usually not exposed at the surface. The extent to which this subterranean water is accessible to herpetofauna is unclear, but we note that we were not able to find common stream-dwelling species (*e.g.,*

*Limnonectes heinrichi*) at the surface above these "underground" streams. Rather, we found tree frogs (*e.g.*, *Chalcorana macrops*) to be abundant above these areas. Despite a significant effort, we only located three springs with running water on the mountain: (1) one small area of exposed water (Lat = 1.06251, Lon = 120.89835) that was part of a larger underground stream just below the 1,000 m camp; (2) a series of small pools fed by another seep (Lat = 1.04917, Lon = 120.91597) at the high camp with small, ~1 m deep pools of tannin-rich, barely moving water; (3) a small stream (Lat = 1.05906 Lon = 120.9364) on the north side of the mountain near the summit which created a few small pools before disappearing below ground. We surmise that the fascinating hydrology of the subterranean streams of Gunung Galang is related to the geology of the Lalos-Toli pluton, on which they are situated.

### Herpetofauna

In total, we collected 638 specimens representing 50 species of amphibians and reptiles over 23 days and several hundred worker-hours of survey effort. In total, we collected 18 species of frogs, 20 species of lizards, 11 species of snakes, and one turtle, *Leucocephalon yuwonoi*. The squamate and amphibian species, numbers of specimens found, and their size, mass, and elevational ranges can be found in Table 1. Appendix A contains color plates of most species. Figure 3 plots all frog, lizard, and snake species arranged by the maximum elevation at which we found specimens and contains elevational species richness curves (assuming range-through) calculated at a 10 m elevational resolution. We were not able to assess the forest structure of Gunung Galang for comparison with *Cannon et al. (2007)* elevationally stratified forest types. We therefore consider elevations up to 700 m to be "low" elevation, elevations from 700–1,400 m to be "middle" elevation, and above 1,400 m to be "high" elevation (Table 2). The 700 and 1,400 m divisions roughly correspond to gaps in our sampling efforts. Our highest-elevation specimens were collected at the peak of Gunung Galang at an elevation of 2,254 m.

We collected 189 specimens representing 28 species during 9 days in the low elevational band, 317 specimens representing 27 species during 17 days in the middle elevational band, and 130 specimens representing 11 species during 11 days in the upper elevational band (Figs. 3 and 4).

We collected four lots of tadpoles representing four species of frogs; *Polypedates iskandari*, *Limnonectes* sp. "T yellow," and *Rhacophorus edentulus*, and *Rhacophorus* sp. "Big Brown".

### Sampling efficacy

Though they differ somewhat, both Chao and Jackknife estimates suggest that we overlooked at least twenty two species in our survey (Table 2). The analyses indicate that one to two frogs, six to eight lizards, and more than a dozen snake species are likely present on Gunung Galang though not detected in our survey. Both Chao and Jackknife estimators indicate that we achieved the most complete sampling in the high elevation band and among amphibians. Our sampling of low elevation environments and of snakes was quite

**Table 1 All frog, lizard, and snake species found during the expedition with corresponding ranges of snout-vent length (SVL), weight, and elevation.**

| Group | Species | n | SVL | Mass | Elevation |
|---|---|---|---|---|---|
| Frog | *Chalcorana macrops* | 31 | 25–50 | 1.17–7.42 | 317–961 |
| Frog | *Chalcorana mocquardi* | 8 | 25–81 | 0.66–23.13 | 300–848.64 |
| Frog | *Ingerophrynus celebensis* | 13 | 26–130 | 1.04–143.54 | 326–1,083 |
| Frog | *Kaloula baleata* | 1 | 56 | 14.62 | 333 |
| Frog | *Limnonectes heinrichi* | 31 | 6.5–78 | 1.49–36.18 | 298–810 |
| Frog | *Limnonectes larvaepartus* | 30 | 11–56 | 0.1–14.35 | 239–1,003 |
| Frog | *Limnonectes modestus* | 2 | 45–47 | 6.95–8.96 | 312–342 |
| Frog | *Limnonectes sp. "T yellow"* | 46 | 24–53 | 1.21–14.53 | 801–1,751.01 |
| Frog | *Occidozyga semipalmata "high elevation"* | 5 | 28–40 | 2–5.85 | 1,732–2,170 |
| Frog | *Occidozyga semipalmata "low elevation"* | 24 | 17–31 | 0.57–3.6 | 326–998 |
| Frog | *Oreophryne "low elevation sp. 1"* | 5 | 20–28 | 0.2–1.88 | 819–995 |
| Frog | *Oreophryne "low elevation sp. 2"* | 2 | 15 | 0.36 | 951–960 |
| Frog | *Oreophryne "middle elevation"* | 36 | 14–27 | 0.28–1.68 | 819–1,770 |
| Frog | *Oreophryne "puncak"* | 11 | 11.5–22 | 0.2–0.92 | 2,170–2,254 |
| Frog | *Papurana celebensis* | 7 | 33–52 | 1.97–7.37 | 326–1,075 |
| Frog | *Polypedates iskandari* | 2 | 55 | 7.76 | 991 |
| Frog | *Rhacophorus edentulus* | 25 | 30–42 | 1.29–3.79 | 788–2,179 |
| Frog | *Rhacophorus sp. "Big Brown"* | 32 | 23–57 | 1–7.28 | 1,720–1,776 |
| Lizard | *Cyrtodactylus fumosus* | 6 | 46.5–98 | 1.45–14.59 | 942–1,213 |
| Lizard | *Cyrtodactylus sp. "pit"* | 23 | 32–66 | 0.56–4.88 | 299–451 |
| Lizard | *Draco spilonotus SNP* | 19 | 44–69 | 1.25–4.67 | 262–1,015 |
| Lizard | *Emoia caeruleocauda* | 6 | 27–50 | 0.39–2.44 | 314–570 |
| Lizard | *Eutropis macrophthalma* | 8 | 94–136 | 21.02–64.07 | 305–1,245 |
| Lizard | *Eutropis multifasciata* | 4 | 57–103 | 5.35–29.38 | 298–450 |
| Lizard | *Eutropis rudis* | 22 | 38–79 | 1.21–14.58 | 285–943 |
| Lizard | *Gekko monarchus* | 1 | 84 | 10.47 | 369 |
| Lizard | *Gekko smithii* | 2 | 168–183 | 78.39–97.63 | 293–298 |
| Lizard | *Hemidactylus frenatus* | 4 | 41–58 | 3.63 | 338–398 |
| Lizard | *Lamprolepis smaragdina "brown"* | 1 | 92 | 13.8 | 293 |
| Lizard | *Sphenomorphus celebensis* | 11 | 36–67 | 1.28–5.38 | 1,447–1,583 |
| Lizard | *Sphenomorphus celebensis "highest"* | 1 | 66.5 | 5.49 | 1,780 |
| Lizard | *Sphenomorphus nigrilabris "high elevation"* | 98 | 25–81 | 0.37–11.49 | 1,012–1,659 |
| Lizard | *Sphenomorphus nigrilabris "low elevation"* | 48 | 23–63 | 0.34–6.27 | 292–1,114 |
| Lizard | *Sphenomorphus sp. "green"* | 2 | 38–81 | 1.02–10.5 | 1,182–1,202 |
| Lizard | *Tytthoscincus "high elevation"* | 11 | 19–45 | 0.16–1.17 | 1,753–1,784 |
| Lizard | *Tytthoscincus "low elevation"* | 17 | 20.5–37 | 0.16–1.02 | 327–451 |
| Lizard | *Tytthoscincus "middle elevation"* | 17 | 28–51 | 0.1–1.33 | 977–1,249 |
| Lizard | *Tytthoscincus "middle orange eye"* | 1 | 35 | 0.84 | 1,067 |
| Snake | *Ahaetulla prasina* | 1 | 1,121 | 133.84 | 220 |
| Snake | *Boiga irregularis* | 1 | 1,066 | 89.1 | 371 |
| Snake | *Calamaria muelleri* | 1 | 151 | 2.32 | 1,038 |

(Continued)

| Group | Species | n | SVL | Mass | Elevation |
|---|---|---|---|---|---|
| Snake | *Calamorhabdium acuticeps* | 3 | 120–173 | 1.54–2.78 | 935–1,015 |
| Snake | *Coelognathus erythrurus* | 1 | 1,219 | 455.9 | 333 |
| Snake | *Cylindrophis melanotus* | 1 | * | * | 363 |
| Snake | *Hebius celebicum* | 1 | 278 | 8.51 | 521 |
| Snake | *Ophiophagus hannah* | 1 | * | * | 774 |
| Snake | *Psammodynastes pulverulentus* | 9 | 187–326 | 2.35–16.49 | 534–1,211 |
| Snake | *Rabdion forsteni* | 1 | 446 | 30.25 | 1,730 |
| Snake | *Rhabdophis chrysargoides* | 3 | 319–510 | 9.18–50.25 | 334–1,137 |

**Notes:**
* Skin shed only.

Note that specimens above ~1,000 m were found in mature, relatively undisturbed forests, whereas specimens below 1,000 m were found in disturbed forest or agricultural land.

incomplete, consistent with our understanding of the additional species present in nearby lowland areas.

## Forest loss

The Gunung Dako Nature Reserve is largely insulated from deforestation, losing just 1.3% of its forest cover since 2000 (*World Resources Institute, 2022c*). Toli-Toli Regency as a whole lost more than 13% of its year-2000 forest cover in the same period, primarily concentrated in lowland areas, especially in valleys south of the city of Tolitoli (*World Resources Institute, 2022b*).

## DISCUSSION

Our survey revealed a diverse herpetofauna on Gunung Galang, with many undescribed species. At least 18 of the 50 species we report here are not clearly referable to existing described species, and many of those remaining are part of diverse species complexes that have radiated across Sulawesi. Substantial taxonomic work is needed to fully understand the true scope of the herpetological diversity present on this mountain. We note that the identities of many specimens were elucidated only after their inclusion in preliminary molecular analyses that included specimens obtained across the entirety of Sulawesi (results not shown). Our results underscore the importance of sustained collection efforts in Sulawesi's middle and high elevations. If species information was only available from lowland sites, not only would many species be overlooked, but the true geographic and climatic distributions of many species would be obscured.

Several genera on Gunung Galang are strikingly elevationally stratified (Fig. 3). For example, we collected five species of the scincid lizard genus *Sphenomorphus* and four species of the scincid genus *Tytthoscincus*, with substantial elevational partitioning. There was only one species present from each of these two genera at the highest (>1,600 m) and lowest (<1,000 m) elevations, but middle elevations held three sympatric *Sphenomorphus* and two sympatric *Tytthoscincus*. A similar degree of elevational stratification was mirrored in the presence of four species of *Oreophryne* (two low, one mid, and one high

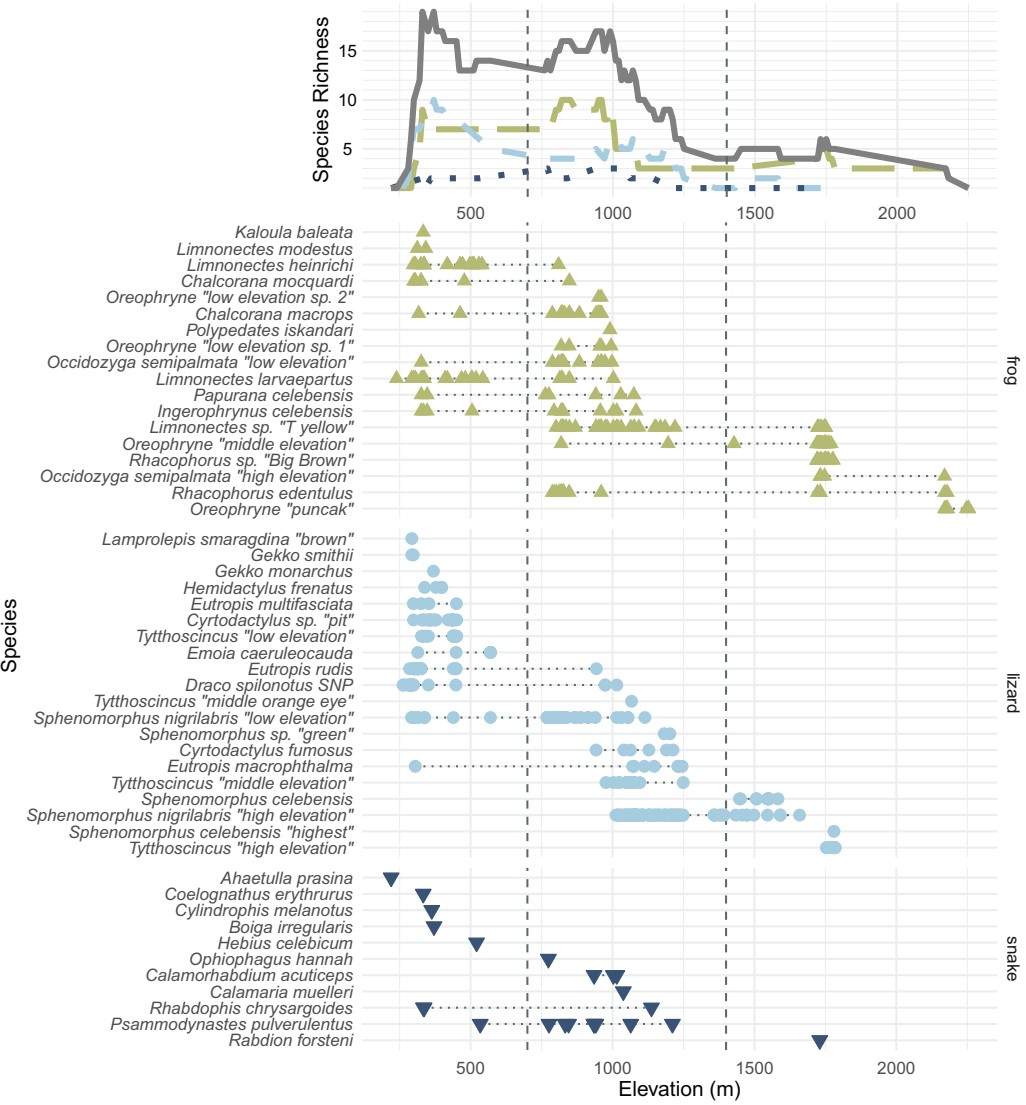

**Figure 3 All frog (green; upright triangles), lizard (light blue; circles), and snake (dark blue; inverted triangles) species and specimens found on Gunung Dako, arranged by maximum elevation.** The upper line plot tallies species richness in 10 m intervals under a range-through assumption. Vertical dashed lines at 700 and 1,400 m demarcate the elevational bands (Table 2). The green, long-dashed line represents frogs; the light blue, short-dashed line represents lizards; the dark blue, dotted line represents snakes. Total richness is represented by the solid gray line.

elevation species—note that these species are given provisional names based on their relative ranges, not the 700 and 1,400 m demarcations between elevational bands), and two lineages of *Occidozyga semipalmata* ("low elevation" and "high elevation"). We suspect that herpetofaunal communities on Gunung Dako are associated with vegetation types, as several species did not occur above or below the stark habitat break we encountered at 1,600 m where the forest shifted from upland hill forest to a mossy montane forest.

We observed a common elevational turnover between many of the lowland and highland species between ~1,000–1,300 m (Fig. 3), with many lowland species' distributed only up to ~700 m. No species were found in all three elevational bands on Gunung

**Table 2 Observed and estimated squamate and amphibian species diversity within taxa and elevational bands.**

| Subset | Individuals found | Observed species | Number of days sampled | Chao estimate | Chao standard error | Second-order Jackknife estimate | Estimated missing species (Chao/Jackknife) |
|---|---|---|---|---|---|---|---|
| Total mountain | 636 | 49 | 23 | 70.52 | 15.42 | 72.69 | 22/24 |
| <700 m | 189 | 28 | 9 | 43.02 | 11.38 | 46.26 | 15/18 |
| 700–1,400 m | 317 | 27 | 17 | 33.02 | 5.61 | 37.45 | 6/10 |
| >1,400 m | 130 | 11 | 11 | 11.91 | 2.09 | 14.45 | 1/3 |
| Frogs | 311 | 18 | 21 | 18.95 | 1.79 | 20 | 1/2 |
| Lizards | 302 | 20 | 22 | 25.97 | 7.23 | 27.59 | 6/8 |
| Snakes | 23 | 11 | 12 | 40.33 | 36.33 | 24.24 | 29/13 |

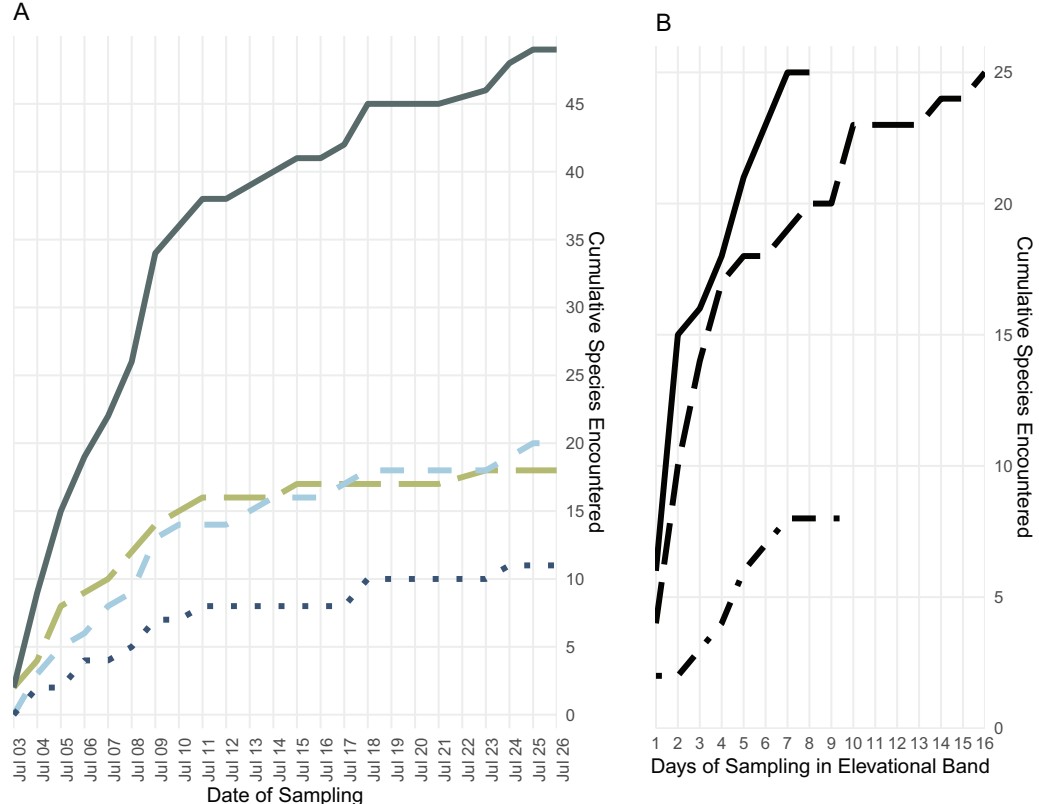

**Figure 4 Species accumulation over the course of the field survey.** (A) Accumulation of frog (green, long dashes), lizard (light blue, medium dashes), and snake (dark blue, dotted) species over the length of the survey period, plus the cumulative species count (gray, solid); (B) accumulation of all frog, lizard, and snake species found below 700 m (solid line), from 700 to 1,400 m (long dashes), and above 1,400 m (dot-dashes) elevation over the number of days during which specimens were collected in that band.

Galang, although *Rhacophorus edentulus* ranged from the 788 to 2,179 m and thus had the largest elevational range of any species in our data set (and undoubtedly occurs lower on the mountain in appropriate habitat). Similarly, *Eutropis macrophthalma* had the largest

predominantly lowland range (305–1,245 m) but we have found this species on other mountains above 2,000 m; we likely simply failed to sample this elusive species at higher elevations.

From our experience conducting fieldwork on Sulawesi, particularly on the Northern Peninsula, there are several species we did not collect that are almost certainly present in lowland areas near the base of Gunung Galang. These include two frog species that we saw in Tolitoli before proceeding to Kinapasan (*Duttaphrynus melanostictus* and *Fejervarya cancrivora*), as well as reptile species that we collected in Kabupaten Tolitoli during an expedition in 2004 (*Gehyra mutilata*, *Dendrelaphis marenae*, *Xenopeltis unicolor*, and *Cuora amboinensis*). Species that we did not collect on this expedition nor in 2004 but that we expect on or near Gunung Galang, based on knowledge of the regional fauna, include the frogs *Hylarana erythraea* and *Rhacophorus georgii*, lizards such as *Lipinia infralineolata*, *Hemidactylus platyurus*, *Lepidodactylus lugubris*, *Bronchocela celebensis*, *Dibamus celebensis*, and *Varanus salvator*, and many snake species such as *Boiga dendrophila*, *Calamaria virgulata*, *Chrysopelea paradisi*, *Gonyosoma jansenii*, *Indotyphlops braminus*, *Malayopython reticulatus*, *Oligodon waandersi*, *Ptyas dipsas*, *Tropidolaemus subannulatus*, and *Xenochrophis trianguligerus*. A few of these species are highly abundant human commensals and/or human introduced, highlighting our relatively low sampling effort in more human-disturbed areas. The Chao and Jackknife estimates for remaining species richness (Table 2) are consistent with these counts of likely unsurveyed lowland species. We note that, in contrast to our sampling of frogs and lizards, our sampling of the snake fauna of Gunung Galang is particularly incomplete, resulting in a high degree of uncertainty around the total snake richness on the mountain (Table 2).

Our sampling of non-snake taxa appears quite comprehensive, especially for anurans. In terms of expected species, we believe we have sampled or observed virtually all of the frog fauna. *Hylarana erythraea* is a rice field species and we did not make an effort to sample rice fields despite their proximity to Gunung Galang. *Rhacophorus georgii* is an extremely rarely-encountered species and appears to be a lowland mature forest obligate, and this habitat type was not available where we worked on Gunung Galang. We failed to find a few lizard species that are likely present where we were sampling, especially the human commensal *Hemidactylus platyurus*, as well as species common in disturbed habitats such as *Lipinia infralineolata* and *Bronchocela celebensis*. *Varanus salvator* is most certainly present, but we pay little attention to this species because it is CITES listed and generally requires specialized traps or snares to collect that we do not employ. *Dibamus* species and *Lepidodactylus lugubris* are never seen or collected in large numbers on Sulawesi. Snakes, of course, are challenging to survey, and even abundant species such as *Chrysopelea paradisi*, *Malayopython reticulatus*, *Tropidolaemus subannulatus*, and *Xenochrophis trianguligerus* are easily missed during a single expedition. Most of the known species that we believe are present but that were not observed are most readily encountered in the lowlands where we had only limited sampling effort in highly disturbed habitats.

Few amphibian or reptile species were observed above 1,800 m on Gunung Galang despite our best attempts to sample the entire transect from ~1,700 m to the 2,254 m
summit using visual surveys, sticky trap arrays, and pitfall traps. Indeed, the only species we collected above 1,784 m were frogs collected at 2,179 m (*Rhacophorus edentulus* and *Occidozyga semipalmata* sp. "high elevation") and *Oreophryne* sp. "puncak" collected between 2,170 m and the 2,254 m summit. This paucity of frogs, and the large sampling gap between 1,784 and 2,179 m clearly indicated in Fig. 1 reflects the lack of accessible free-flowing water between our ~1,700 m high camp and the small stream near the summit. Our inability to find reptiles above 1,784 m is unlikely to be a simple matter of sampling effort, as we established sticky trap lines that extended from 1,920–2,254 m, as well as pitfall arrays at 2,170 m. Rather, we conclude that few reptiles occur above 1,800 m on Gunung Galang. Our species accumulation curves are consistent with this interpretation (Fig. 3). The sampling curve for elevations 1,400 m and above reached a plateau after 8 days of sampling (Fig. 4), but the rate of species accumulation in this band was slow enough that there could be a few remaining species. Given this, and the relative accuracy of the Chao and Jackknife estimates for remaining lowland species richness, we believe that three or more additional species may exist above 1,600 m. These remaining unsurveyed species would likely represent undescribed species, as most others found in this zone are currently undescribed. Even after considering unsurveyed species, the highland fauna on Gunung Galang and larger Gunung Dako area is relatively depauperate when compared to highland areas on the adjacent Sunda Shelf, where faunas at all elevations are admittedly much richer than on Sulawesi. For example, Gunung Kinabalu on Borneo, one of the most biodiverse mountains in all of Southeast Asia, holds 28 frog species, nine snake species, and four lizard species at 2,000 m elevation (*Malkmus, 2002*). This is likely partially attributable to the much richer source fauna on Borneo (*McGuire et al., 2023*), and the fact that there is vastly more habitat available above 2,000 m on the 4,095 m Gunung Kinabalu than is present on the much smaller Gunung Galang.

On tropical mountains, species richness in reptiles and amphibians often either peaks at middle elevations or holds a plateau between low and middle elevations before declining at higher elevations (*McCain & Grytnes, 2010*). For example, in South American frogs, a mid-elevation richness peak is likely due to the long geographic residence of anurans and increased diversification rates in the Andes (*Hutter, Lambert & Wiens, 2017*). After accounting for the lowland species not encountered on this survey, we find that species richness of snakes and lizards is highest at low elevations, declining at middle elevations and further declining at high elevations, a pattern similar to that of Gunung Galang's ferns (*Susila, Jamhari & Kasim, 2020*). For frogs, however, we observed a plateau in richness between low and middle elevations, with declining richness in high elevations. Since the mountains of the Northern Peninsula uplifted only a few million years ago, it is possible that these middle and high elevations are still in the process of accumulating faunal diversity. In addition, while 13 of 28 lowland species were found at middle elevations, only 4 of 27 middle-elevation species were found at high elevation, so the expanded elevational ranges of lowland species could be driving this elevational diversity gradient.

Our analysis of forest loss within the Gunung Dako Nature Reserve and Tolitoli Regency indicates that most forest loss is restricted to low and middle elevation agricultural development (*World Resources Institute 2022b*, *2022c*). In the nature reserve,

loss was limited to the western slope adjacent to the neighboring villages below 1,000 m. Middle and high elevation environments did not experience substantial forest loss over the past 10 years. This indicates that the forest reserve is successfully protecting species in middle and high elevation forests that might otherwise be logged. Lowland forest has not experienced substantial recent loss, mirroring patterns across Sulawesi (*Margono et al., 2014*) likely because it was logged many years ago for agriculture. The steep and rugged terrain and limited stable bedrock in these mountain ranges may inherently protect middle and high elevation environments from extensive logging. Selective logging along trail systems is unlikely to pose a major conservation threat.

Despite these encouraging trends, Gunung Dako's forests are not insulated from human impacts. The social restrictions, impoverishment, and cessation of community programs brought on by the COVID-19 pandemic have forced some locals into unsustainably extractive activities in the Gunung Dako region (*Golar et al., 2020*). So long as poverty exists in Toli-Toli and the areas surrounding Gunung Dako, the reserve's forests are at risk.

Despite the massive efforts undertaken, we are just beginning to understand the biodiversity of Gunung Galang and the Gunung Dako Nature Reserve. Records of the birds, mammals, and invertebrates collected on this expedition have yet to be fully analyzed, and there has been little systematic sampling of the flora of the Gunung Dako Nature Reserve (*Susila, Jamhari & Kasim, 2020*). Considering the area's complex tectonic history (*Rahbek et al., 2019a*), we consider it likely that the reserve's flora and arthropod fauna are just as diverse and endemic as its herpetofauna. Sulawesi's mountains may host hundreds or thousands of species yet unknown to science.

## ACKNOWLEDGEMENTS

We are grateful for the critical help and support of our larger field expedition team for making this project not only possible, but enjoyable. In particular, we thank Anang Achmadi, Mohammad Irham, and Pungki Lupiyaningdya for coordinating the expedition, and Dede Avandi for his help managing permits and visas. There are too many members of our expedition to name everyone, but we are also particularly thankful to Donny Bosco for working to establish the camps and Safir for cooking excellent field meals for our large team and always keeping us laughing. We further thank the people of the village of Kinopasan for their support and assistance in making this expedition possible, including housing us, carrying thousands of pounds of gear and equipment up the mountain, and their assistance making exceptional field camps under the particularly difficult conditions we faced. We also thank Carol Spencer and her curatorial team for their help importing specimens and accessioning them in the MVZ collection.

### Funding

This research was supported by the National Science Foundation of the United States (No. 1457845). The funders had no role in study design, data collection and analysis, decision to publish, or preparation of the manuscript.

## Grant Disclosures

The following grant information was disclosed by the authors:
National Science Foundation of the United States: 1457845.

## Competing Interests

The authors declare that they have no competing interests.

## Author Contributions

- Benjamin R. Karin conceived and designed the experiments, performed the experiments, analyzed the data, prepared figures and/or tables, authored or reviewed drafts of the article, and approved the final draft.
- Isaac W. Krone conceived and designed the experiments, performed the experiments, analyzed the data, prepared figures and/or tables, authored or reviewed drafts of the article, and approved the final draft.
- Jeffrey Frederick conceived and designed the experiments, performed the experiments, analyzed the data, prepared figures and/or tables, authored or reviewed drafts of the article, and approved the final draft.
- Amir Hamidy conceived and designed the experiments, performed the experiments, authored or reviewed drafts of the article, and approved the final draft.
- Wahyu Tri Laksono conceived and designed the experiments, performed the experiments, authored or reviewed drafts of the article, and approved the final draft.
- Sina S. Amini performed the experiments, authored or reviewed drafts of the article, and approved the final draft.
- Evy Arida conceived and designed the experiments, performed the experiments, authored or reviewed drafts of the article, and approved the final draft.
- Umilaela Arifin performed the experiments, analyzed the data, authored or reviewed drafts of the article, and approved the final draft.
- Bryan H. Bach performed the experiments, authored or reviewed drafts of the article, and approved the final draft.
- Collin Bos performed the experiments, authored or reviewed drafts of the article, and approved the final draft.
- Charlotte K. Jennings performed the experiments, authored or reviewed drafts of the article, and approved the final draft.
- Awal Riyanto conceived and designed the experiments, performed the experiments, authored or reviewed drafts of the article, and approved the final draft.
- Simon G. Scarpetta performed the experiments, authored or reviewed drafts of the article, and approved the final draft.
- Alexander L. Stubbs performed the experiments, authored or reviewed drafts of the article, and approved the final draft.
- Jimmy A. McGuire conceived and designed the experiments, performed the experiments, analyzed the data, prepared figures and/or tables, authored or reviewed drafts of the article, and approved the final draft.

## Animal Ethics

The following information was supplied relating to ethical approvals (*i.e.*, approving body and any reference numbers):

The animal study protocol was approved by the Institutional Animal Care and Use Committee of the University of California Berkeley.

## Field Study Permissions

The following information was supplied relating to field study approvals (*i.e.*, approving body and any reference numbers):

Research and collection permits were granted by LIPI and RISTEKDIKTI (now BRIN).

## Data Availability

All data and code are available at Zenodo: dibamus. (2023). dibamus/ Gunung_Dako_Checklist: Submission to PeerJ (1.1). Zenodo. https://doi.org/10.5281/ zenodo.7999977.

## Supplemental Information

Supplemental information for this article can be found online at http://dx.doi.org/10.7717/ peerj.15766#supplemental-information.

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
