# Peer review of "Elevational surveys of Sulawesi herpetofauna 1: Gunung Galang, Gunung Dako Nature Reserve"

_PeerJ, doi:10.7717/peerj.15766_

## Round 0.1 · original submission · Minor Revisions

Dear authors,

After undergoing three revisions, the reviewers have commended the manuscript for its exceptional quality and insightful analyses, which effectively showcase your expertise in the subject matter and the field of study. Consequently, only minor corrections are required, particularly in certain figures. Additionally, it is advisable to separate the conclusions from the discussions and attend to other minor details.

We greatly appreciate your invaluable efforts in preparing this manuscript, and we eagerly await your feedback.

Best regards,

Armando Sunny

·

Basic reporting

The submitted manuscript aims to provide a review and reporting of the survey efforts done on Gunung Galang of Sulawesi. The authors provide an incredibly thorough report and analysis of the encountered herpetofauna across elevational gradients. I have minor edits, which are primarily just mere suggestions or grammar corrections. The figures look professional, and only minor changes are suggested (see attached file). Additionally, I greatly appreciate the authors' submission of a Bahasa Indonesia abstract.

Experimental design

I commend the authors for their meticulous detail and reporting on habitat, habitat loss, and elevation data using repeatable methods. The research is original and is much-needed for understudied regions, including many islands in Southeast Asia. Thus, this fills in many gaps we have (and still have) about island diversity in such regions. The methods included several survey and collections methods that are commonly employed that are widely used today (pitfall traps, sticky traps, opportunistic sampling).

Validity of the findings

All findings and conclusions are valid, and are based on robust collection efforts and sound methods.

Additional comments

My only additional comment is that I did not see a reference to Supplemental Material in the text, so perhaps that can be added at some point in the manuscript.

·

Basic reporting

This is a well-written manuscript that was a pleasure to read and I have no issues with the language as many of the authors are proficient English speakers. The authors are also very experienced in working in that region and have conducted decades of work its herpetofauna. They did a very thorough coverage of the literature of the focus location in the Introduction and all the pertinent literature was referenced. The paper was concisely structured and the results were presented clearly in the paper or available as supplementary material.

Experimental design

The methods and analyses used were all relevant to the study and satisfactory. The team conducted very thorough fieldwork for almost a month at the mountain and were meticulous is covering different elevations and employing a variety of sampling techniques from visual encounter surveys to pitfall and sticky traps to to maximize their findings. Well done!

Validity of the findings

Data are all generally well presented. I only have a minor comment in that the number of species reported in the text is not matching up with that presented in the tables. You report in the text a total of 50 species but based on the taxa listed in the tables, there should only be 49. I have annotated my comments on the attached pdf.

Additional comments

Recheck the supplemental figures for the species. There are two frogs labelled as Limnonectes sp. "T yellow". The second image appears to be a species of Oreophryne.

Also double check the references and citations in text for lines 377-378.

All my comments have also been annotated onto the pdf.

Reviewer 3 ·

Basic reporting

I enjoyed reading the manuscript entitled “Elevational surveys of Sulawesi herpetofauna 1: Gunung Galang, Gunung Dako Nature Reserve” very much. It is very nice to know the differences in herpetofauna composition across different elevational bands in one of Sulawesi’s Mountain. Overall, it is written very clearly. That said, I do have several comments which could be given consideration to refine the manuscript:

1. The article is sometimes written in the form of a popular press article, and could be streamlined with less personal references. For example, In line 235, 237–238.
2. There are some inconsistencies in formatting, such as not using an en dash for all elevational ranges, using space and no space between number and unit, the number of collected specimens, also in the caption of figures.
3. In many places in the manuscript the authors use the term “diversity” but given the context I think they probably mean “richness”. Please check all usages of ‘diversity”.
4. Figure 4B is not really clearly explained (see my comments on the figure).
5. It seems the authors combine discussion and conclusion together. Maybe authors can consider to separate them.

I have provided additional comments and edits on the annotated pdf (please note each highlighted sentence has a specific comment), which is attached with this review.

Experimental design

no comment

Validity of the findings

no comment

Annotated reviews are not available for download in order to protect the identity of reviewers who chose to remain anonymous.

---

## Round 0.2 · accepted · Accept

Dear authors,

We are pleased to inform you that, following careful consideration of the comments provided by the reviewers, we have reached a consensus that the previous observations made in your manuscript were indeed accurate. Consequently, we are delighted to accept your work for publication.

We extend our sincerest gratitude for your dedication and hard work in conducting this intriguing research.

Best regards,

Armando Sunny

Reviewer 3 ·

Basic reporting

no comment

Experimental design

no comment

Validity of the findings

no comment

Additional comments

Authors have addressed previous concerns very well. I am satisfied with the revised version.